# Uni-YOLO: Vision-Language Model-Guided YOLO for Robust and Fast Universal Detection in the Open World

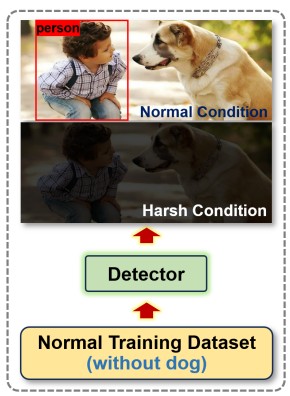 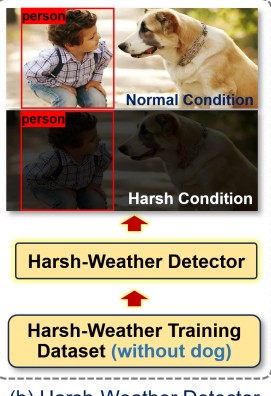 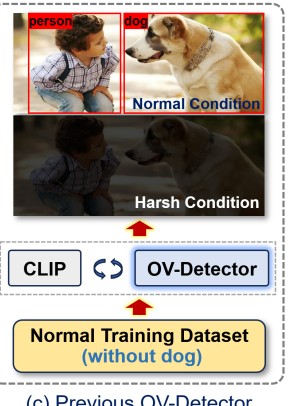 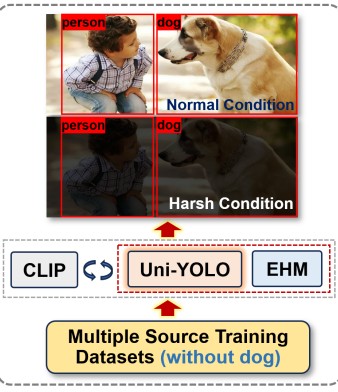

(a) Traditional Detector   (b) Harsh-Weather Detector   (c) Previous OV-Detector   (d) Our Uni-YOLO

**Figure 1: Illustration of four object detectors. (a) Traditional detector: detects only the categories in its training dataset under normal conditions. (b) Harsh-Weather Detector: detects only the categories in its training dataset under normal or harsh conditions. (c) Open Vocabulary Detector: detects the categories that are not present in its training dataset under normal conditions. (d) Our Uni-YOLO: detects the categories that are not present in its training dataset under normal or harsh conditions. It utilizes multiple source datasets for better generalization and uses EHM for harsh weather robustness.**

## ABSTRACT

Universal object detectors aim to detect any object in any scene without human annotation, exhibiting superior generalization. However, the current universal object detectors show degraded performance in harsh weather, and their insufficient real-time capabilities limit their application. In this paper, we present Uni-YOLO, a universal detector designed for complex scenes with real-time performance. Uni-YOLO is a one-stage object detector that uses general object confidence to distinguish between objects and backgrounds, and employs a grid cell regression method for real-time detection. To improve its robustness in harsh weather conditions, the input of Uni-YOLO is adaptively enhanced with a physical model-based enhancement module. During training and inference, Uni-YOLO is guided by the extensive knowledge of the vision-language model CLIP. An object augmentation method is proposed to improve generalization in training by utilizing multiple source datasets with heterogeneous annotations. Furthermore, an online self-enhancement method is proposed to allow Uni-YOLO to further focus on specific objects through self-supervised fine-tuning in a given scene. Extensive experiments on public benchmarks and a UAV deployment are conducted to validate its superiority and practical value.

## CCS CONCEPTS

• **Computing methodologies → Object detection**.

## KEYWORDS

object detection, zero-shot learning, vision-language model, CLIP

## 1 INTRODUCTION

The dependence on human annotations and the numerous categories present in the open world significantly limit the universality of traditional object detectors. In complex and variable environments, it is unfeasible to collect and annotate all data for every scene [6, 27, 28, 53, 56]. To address these data limitations, a universal visual object detector is necessary [44]. The universal detector aims to detect any object (open vocabulary) in any scene (open world) and can refine itself in a new scene without human annotations.

Recently, some large language models (LLMs), such as GPT [1], and ERNIE [40], demonstrate superior generalization performance in natural language processing. Researchers are now exploring how to extend the generalization capabilities of LLMs to visual models. Some large-scale vision-language pre-training models, such as CLIP [31], have been proposed. In the field of object detection, open vocabulary detectors (ov-detectors) [10, 17, 26, 44, 51, 57] use CLIP [31] to recognize unknown categories. However, these current ov-detectors face limitations in addressing two important aspects, which restrict their universality. *1) The universal detector should have better generalization and robustness to detect objects in the open world.* Although some novel categories can be detected by existing ov-detectors, detecting a wide range of unknown categories in the open world remains a challenge [44]. Moreover, current ov-detectors

exhibit poor robustness in harsh weather conditions, and real-world scenes are inevitably affected by factors such as scattering and low illuminance, which lead to unsatisfactory generalization of existing detectors. *2) The universal detector should also have better real-time performance to enable deployment on mobile platforms.* Most existing ov-detectors rely on the Faster RCNN architecture [35], which has a two-stage architecture that prioritizes detection accuracy in a closed set, albeit at the expense of real-time performance. A more efficient architecture for universal detectors will facilitate deployment on mobile platforms and enhance the practical application value. Thus, unlike ov-detectors, a universal detector should have better open-world generalization, robustness, and efficient architecture.

To achieve a universal detector, we focus on further improving open-world generalization, and we need to address the following three technical challenges. *1) How to generalize to the complex open world without category supervision.* Given the abundance and variety of object categories, it is not feasible to provide complete annotation for each object in training. Additionally, the open-world environment is complex and variable, and the imaging process by vision sensors is susceptible to various harsh weather conditions, resulting in less robust detection. We propose general object confidence to directly learn the general features of all objects, and enhance object features based on physical imaging models in harsh weather conditions. *2) How to train a universal detector using multiple source datasets with heterogeneous annotations.* A universal detector should be trained on large datasets for better generalization. However, existing datasets such as COCO [21] and Object365 [36] are based on different human annotation criteria, resulting in cases where some objects may be annotated as background in another dataset. Such inconsistent annotation prevents the detector from learning the general features of objects. We propose an object augmentation method to generate consistent annotation. *3) How to adapt and improve itself in a new scene without human annotation.* At the core of intelligence is adaptation and learning, as if even a child can generalize rapidly in a new environment [44]. When a well-trained universal detector is applied to a given scene, the categories of objects to be detected are usually also given. In this case, the universal detector should be able to adapt itself to improve the detection of given categories while minimizing the focus on irrelevant categories. We propose a self-enhancement method to further improve the detection of specific objects in a given scene.

In this paper, we propose Uni-YOLO, a robust universal object detector for the complex open world with real-time performance. Unlike most existing ov-detectors, Uni-YOLO is designed as a one-stage detector. The input of Uni-YOLO is enhanced with a physical model-based enhancement module (EHM) to provide adaptive enhancement for various complex weather conditions, rather than directly using the original degraded images as in previous methods. To detect more unknown objects, we propose a General Object Confidence (GOC). Based on GOC, Uni-YOLO learns the general features of objects to effectively discriminate between numerous categories and backgrounds. For zero-shot classification, Uni-YOLO is designed with a Matching Head to perform contrastive classification with the text embeddings of candidate objects. During training and inference, Uni-YOLO is guided by the extensive knowledge of the vision-language model CLIP. An object augmentation method is proposed to achieve consistent annotation for multiple source

datasets. Based on generalization training with multiple source datasets, Uni-YOLO learns the general features of innumerable objects to achieve better generalization. To further improve the detection in a given scene, an online self-enhancement method is proposed. Uni-YOLO assigns pseudo-labels exclusively to given objects and performs fine-tuning based on these labels. Based on Uni-YOLO, we develop a UAV platform for multimedia interaction detection. It is demonstrated that Uni-YOLO can maintain real-time detection based on a low computational platform in the open world.

The main contributions of this work are summarized as follows:

- A new one-stage universal detector named Uni-YOLO is proposed. It includes three detection Heads operating in parallel to perform contrastive classification with the text embeddings of candidates, and uses a physical model-based EHM to improve its robustness in harsh weather conditions.
- For training Uni-YOLO, an object augmentation training method is proposed to achieve better zero-shot generalization. The method addresses the heterogeneous problem of multiple source datasets and achieves large-scale training.
- For online fine-tuning Uni-YOLO, a self-enhancement method is proposed. The method enables the detector to improve the detection of given objects in any given scene.

Extensive experiments are conducted to validate the superiority of Uni-YOLO on various public object detection benchmarks.

## 2 RELATED WORK

### 2.1 Traditional Object Detection

Object detection tasks, involving object classification and localization, are crucial in computer vision. Current learning-based detectors can be broadly classified into three categories: two-stage methods, one-stage methods, and transformer-based methods. Two-stage detectors first extract a set of region proposals and then perform classification and regression, such as Faster-RCNN [35]. One-stage detectors perform classification and localization directly on the input images. One-stage detectors, especially YOLO [34], exhibit remarkable real-time detection performance. Transformer-based detectors, such as DETR [2] and others [4, 22, 52], are rapidly evolving. However, these traditional detectors only work on a closed set [9, 19, 48], which requires a lot of human annotations for training.

### 2.2 Open-Vocabulary Object Detection

Open-vocabulary detectors aim to detect categories that do not appear in the training dataset. Various open-set object detection methods [23, 27, 33, 39, 59, 60] are proposed to generalize to unknown categories from the base categories. For example, Liang *et al*. propose UnSniffer [20], which uses two detector heads for both base and unknown categories and introduces general object confidence to achieve unknown localization. However, these methods provide only the localization, but not the classification for unknown categories. Subsequently, the advent of large-scale vision-language pre-training models has led to advances in ov-detectors [10, 17, 26, 37, 44, 51, 57]. The ov-detectors use vision-language models to achieve classification for novel categories. Zhong *et al*. propose RegionCLIP [57], an extension of CLIP, to learn regional visual information for finer-grained alignment of images and text. Wang *et al*. propose a universal object detector, named UniDetector

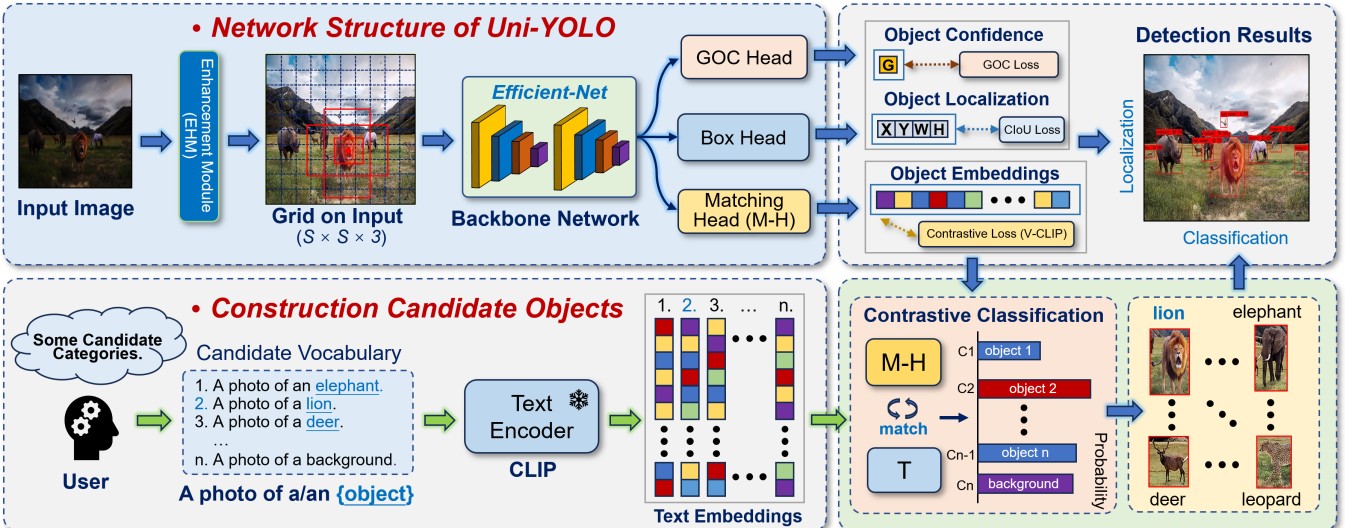

**Figure 2: Illustration for the pipeline of our Uni-YOLO. Uni-YOLO first performs adaptive enhancement for the input degraded image and divides the enhanced image into an $S \times S$ grid cell. Each grid cell provides three sizes of a priori boxes and is responsible for predicting the objects in its center. Uni-YOLO uses the Box Head and the GOC Head to perform bounding box regression, and uses the Matching Head to perform contrastive classification for candidate objects.**

[44], with the ability to detect a wide range of object categories in the open world. It is designed by a unified structure to use multiple sources of datasets for localization training and it employs text embeddings for classification. However, these existing ov-detectors are designed only for normal environments. Natural environments in the open world are inevitably affected by harsh weather, leading to unsatisfactory practical performance of existing ov-detectors.

### 2.3 Object Detection in Harsh Environments

Most methods first design an image enhancement network to enhance degraded images and then perform object detection based on these enhanced images [3, 8, 30, 38, 41, 42, 49, 54]. However, these enhancement modules are originally designed for human vision, as measured by image quality metrics, and not for detection accuracy [16, 24]. For this reason, some studies introduce enhancement methods that are specifically tailored to object detection tasks. IA-YOLO [24] proposes a trainable image processor and uses detection loss training to improve object detection performance. BAD-Net [16] uses an attention fusion module to combine the features of the original images with those of the enhanced images, assuming that the original images contain valuable information for detection. 2PCNet [12] introduces a two-stage consistent unsupervised domain-adaptive network and applies domain-adaptive methods to enable detection in low illuminance conditions. However, these enhancement and detection methods work only in a closed set and cannot detect any unknown object in an open-world scene.

## 3 METHOD

### 3.1 The Pipeline of Uni-YOLO

The pipeline of Uni-YOLO is illustrated in Figure 2. Uni-YOLO is designed as a new one-stage architecture with three detection heads operating in parallel. The detection results are based on the localization coordinates provided by the General Object Confidence

Head (GOC Head) and the Box Regression Head (Box Head), and the classification results provided by the Matching Head.

**Object Localization:** We utilize Efficient-Net [43] as the backbone network for feature extraction. Uni-YOLO divides the input image into $S \times S$ grid cells, and each grid cell is responsible for detecting objects that fall within its center. Each grid cell predicts three sizes of bounding boxes $(x, y, w, h)$ provided by the Box Head, which represent the location of the bounding boxes (center coordinates, width, and height). Each bounding box has a corresponding general object confidence provided by the GOC Head. We determine whether there is any object in each candidate bounding box by a GOC threshold and output the localization coordinates.

**Object Classification:** Uni-YOLO achieves zero-shot classification through the Matching Head. Each bounding box has a corresponding object embedding provided by the Matching Head, denoted as $V = [\Psi_V(box_1), \Psi_V(box_2), ..., \Psi_V(box_j)]$. Uni-YOLO uses CLIP's text encoder to generate text embeddings for retrieving candidate objects. Specifically, we construct a vocabulary list of objects that are of interest to the user for the candidates, such as $List = [person, lion, ..., elephant, |background, object]$[1]. We use a prompt template, i.e., "A photo of a/an object," and feed the prompts into the text encoder to obtain a set of text embeddings $L = [\Psi_L(person), \Psi_L(lion), ..., \Psi_L(elephant)]$. If a bounding box contains any object (GOC above threshold), we perform similarity matching in the image-text space to classify the object corresponding to the maximum similarity between the text embeddings and features, thus achieving zero-shot classification for any object:

$$p_{ij} = Maximum(SoftMax(Sim(V[j], L[i]))), \quad (1)$$

where $p_{ij}$ denotes the probability that the $j$th bounding box belongs to the $i$th candidate category. If the candidate corresponding to

---

[1]Different from the candidates of the traditional CLIP zero-shot classification method [31], our list introduces two additional candidates, "background" and "object", to mitigate the potential misclassification of the background region.

the maximum probability is "background" or "object," the candidate box is discarded for further classification corrections.

## 3.2 The Architecture of Uni-YOLO

### 3.2.1 General Object Confidence.
Given the abundance and diversity of object categories, providing specific and consistent descriptions for the features of each object is challenging. Although humans struggle to provide specific descriptions, we can still identify overarching distinguishing features of objects and backgrounds; for example, there is usually an obvious boundary between them. It is suggested that the detector can also learn the overarching features from a large dataset spanning various categories.

We propose the General Object Confidence (GOC) to discriminate whether a bounding box contains objects. The GOC is defined as the prediction results of the detection head based on the RepConv structure [43], denoted as $\Phi(b_i)$. The range of GOC is $[0, 1]$, with a higher value indicating a higher probability that the bounding box contains an object. In inference, we filter all bounding boxes based on a GOC threshold to identify those that may contain potential objects. In training, we design three losses to train GOC. The design of these losses takes into account three practical situations aimed at detecting more objects with better generalization.

**Case 1: Complete Confidence Objects.** This case contains complete confidence objects (manually annotated boxes in training datasets), and their corresponding bounding box should have a GOC value of 1. Thus, the first GOC loss is expressed as:

$$L_{cco} = \frac{1}{N} \sum_{i \in [1,N]} \frac{1}{|B_{cco}|} \sum_{b_i \in B_{cco}} (\hat{\Phi}(b_i) - 1)^2, \quad (2)$$

where $\hat{\Phi}(b_i)$ represents the predicted probability of GOC in the proposal bounding boxes $b_i$. $B_{cco}$ is the set of complete confidence annotations. $N$ is the number of proposal bounding boxes.

**Case 2: Contrastive Confidence.** This case involves the contrastive confidence between bounding boxes, i.e., the more precise the predicted localization of the proposals, the higher the value of their GOC. Thus, the second GOC loss is expressed as:

$$L_{co} = \frac{1}{N} \sum_{i \in [1,N]} \frac{2}{|B_{cco}|} \sum_{b_j, b_k \in B_{cco}} max(\frac{\hat{\Phi}(b_j) - \hat{\Phi}(b_k)}{\alpha} + \zeta, 0), \quad (3)$$

where $\alpha = 1$ if $IoU(\hat{\Phi}(b_k), y_i) > IoU(\hat{\Phi}(b_j), y_i)$; otherwise, $\alpha = -1$, $y_i$ is the annotation. $\zeta$ is a tiny constant that set to 0.01.

**Case 3: Complete Confidence Background.** This case involves a complete confidence background (discriminated as background by the CLIP), and its corresponding bounding box should have a GOC value of 0. Thus, the third GOC loss is expressed as:

$$L_{ccb} = \frac{1}{N} \sum_{i \in [1,N]} \frac{1}{|B_{ccb}|} \sum_{b_i \in B_{ccb}} (\hat{\Phi}(b_i) - 0)^2, \quad (4)$$

where $B_{ccb}$ represents the set of complete confidence backgrounds, determined based on the proposed object augmentation training method (subsection 3.3). Thus, the total GOC loss is:

$$Loss_{goc} = L_{cco} + L_{co} + L_{ccb}. \quad (5)$$

Additionally, we use a regression loss, denoted as $Loss_r$, to perform bounding box regression training. The regression loss regresses all bounding boxes containing potential objects to achieve more accurate localization, and it can be roughly expressed as:

$$Loss_r = 1 - CIoU(b_i, label_i), \quad (6)$$

where $CIoU(\cdot)$ represents the Complete-IOU, as proposed by [55].

### 3.2.2 Contrastive Classification.
We design a contrastive classification loss to train the Matching Head. It is designed to compute the similarity between the proposal bounding boxes containing potential objects and the object embeddings of the annotated regions extracted by the vision encoder of CLIP. It is expressed as:

$$Loss_{con} = \frac{1}{N} \sum_{i \in [1,N]} \frac{1}{|An|} \sum_{n_j \in An} (1 - Sim(\Psi_V(box_i), V_{clip}(n_j))), \quad (7)$$

where $An$ is the set of annotated object regions. $Sim(\cdot)$ is the cosine similarity. The contrastive classification loss represents the knowledge transfer from the CLIP model to the Matching Head.

### 3.2.3 Enhancement Module.
We design a physical model-based enhancement module (EHM) to perform image enhancement in harsh weather conditions for more robust detection. We mainly consider two common degradations: scattering and low illuminance.

**Scattering Degradation.** The atmospheric scattering model [29] is used to describe the degradation in scattering environments:

$$J(x) = \frac{1}{t(x)}I(x) - A(x)\frac{1}{t(x)} + A(x), \quad (8)$$

where $I(x)$ is the degraded scattering image, and $J(x)$ is the enhanced image. $t(x) = e^{-\beta d(x)}$ is the medium transmission map, where $d(x)$ is the scene depth and $\beta$ is the scattering density scattering coefficient. $A(x)$ is the global atmospheric light. Traditional methods use two networks, represented $\phi[I(x)]$ and $\psi[I(x)]$, to estimate $t(x)$ and $A(x)$. The enhanced image is computed as follows:

$$J(x) = \frac{1}{\phi[I(x)]}I(x) - \psi[I(x)]\frac{1}{\phi[I(x)]} + \psi[I(x)]. \quad (9)$$

To reduce the computational complexity of two parameter estimation, we combine the parameters, $t(x)$ and $A(x)$, into a dehazing map $D_m(x)$, by referring [14]. The reformulated model is:

$$J(x) = D_m(x)(I(x) - 1) + 1, \quad (10)$$

$$D_m(x) = \frac{\frac{1}{t(x)}(I(x) - A(x)) + (A(x) - 1)}{I(x) - 1}. \quad (11)$$

Thus, we can use one network to estimate the dehazing map $D_m(x)$ to perform enhancement in scattering environments.

**Illuminance Degradation.** We design a learnable *Gamma* corrector to achieve enhance for low illuminance conditions. The *Gamma* corrector improves the contrast of degraded images through a nonlinear transformation, defined as follows:

$$J_o(x) = J(x)^{\gamma(x)} = (r_i(x)^{\gamma_r(x)}, g_i(x)^{\gamma_g(x)}, b_i(x)^{\gamma_b(x)}), \quad (12)$$

where $\gamma_{r,g,b}(x)$ is the correction map and different values correspond to different mappings for illuminance degraded images. We also use one network to estimate the correction map $\gamma_{r,g,b}(x)$ to perform enhancement in low illuminance environments.

**Enhancement Network.** For real-time detection performance, we design a lightweight two-branch network, denoted as $\phi_{D_m, \gamma}[I(x)]$,

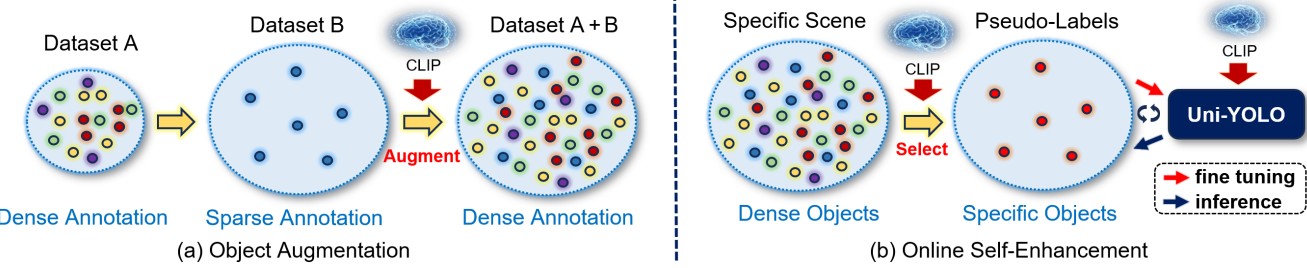

**Figure 3: Illustration for the EHM. The degraded image is firstly extracted features through a dense backbone and then passes through a two-branch structure to obtain the dehazing map and correction map. Finally, the enhancement is performed based on physical models. The EHM is pre-trained based on the perceptual loss [11], and then is jointly trained with our detector.**

**Figure 4: Illustration for the object augmentation and self-enhancement methods. (a) Object Augmentation. Under the guidance of the vision-language model (CLIP), the sparse annotation dataset is transformed into a dense consistent annotation dataset, to achieve large-scale training using multiple source datasets with heterogeneous annotation. (b) Online Self-Enhancement. Under the guidance of CLIP, Uni-YOLO performs self-enhancement for specific objects in any given scene.**

to parallelly obtain the dehazing map $D_m(x)$ and the correction map $\gamma_{r,g,b}(x)$ for degraded image adaptive enhancement:

$$J_s(x) = \phi_{D_m}[I(x)](I(x) - 1) + 1,$$
$$J_g(x) = J_s(x)^{\phi_\gamma[I(x)]}. \tag{13}$$

The two-branch network allows the two sets of parameters to be estimated using one single backbone. The specifics of the designed two-branch network are shown in Figure 3, and more parameter and training details are provided in our supplementary material.

### 3.3 Object Augmentation Training Method

The generalization performance of large models improves with the increased availability of training data [44]. Ensuring that Uni-YOLO is trained to its full potential using existing datasets is crucial for its generalization. However, the challenge arises from the heterogeneous annotations present in multiple source datasets. For example, in the COCO dataset [21], the object "book" is annotated as an object, while in the Pascal VOC dataset [5], it is ignored as background. This inconsistency hampers the detector's ability to learn the overarching features of general objects when simply merging multiple source datasets for large-scale training.

To address this inconsistency, we propose an object augmentation method to achieve consistent annotations across multiple source datasets for Uni-YOLO training. As shown in the left part of Figure 4, we first train our Uni-YOLO using the most densely annotated dataset. The pre-trained Uni-YOLO is then used to perform inference on a sparser dataset for obtaining more annotations, with the correctness of the pseudo-labels determined under the guidance

---

**Algorithm 1** Object Augmentation Training Method

**Input:** We only use the location annotations: $label = \{x, y, w, h\}$ in the training datasets: $\{D_1, D_2, .., D_n\}$. And the output proposal bounding box of Uni-YOLO is denoted as $\Theta(image)$.
1: Train $\Theta(image)$ with relatively densest annotation dataset $D_m$.
2: Construct the training database $DS = \{D_m\}$.
3: Construct the candidates $List = [object, background]$.
4: **for** $i$ in $\{D_1, D_2, .., D_n\}$ **do**
5:    **for** $j$ in $D_i$ **do**
6:       $[image(ij), label(ij)] = D_{ij}$.
7:       $embedding(ij) = \Theta(image(ij))$.
8:       $p_k = SoftMax(sim(L(List[k]), V(embedding(ij))))$.
9:       $pseudo\_label(ij) = List[Max(p_k)]$.
10:      $label(ij) = label(ij) \cup pseudo\_label(ij)$.
11:    **end for**
12:    Update $DS = \{DS, D_i\}$.
13:    Update $\Theta(image)$ with $Loss_{goc}$, $Loss_r$ and $Loss_{con}$ on $DS$.
14: **end for**
**Output:** The trained $\Theta(image)$. And given any image, it has the ability to provide detection for all objects.

---

of the large vision-language model CLIP. The pseudo-labels, along with the original annotations, form more dense and consistent annotations, and Uni-YOLO is retrained on the augmented datasets. The process is applied iteratively to multiple datasets for training. The specific steps are summarized in Algorithm 1.

---

**Algorithm 2** Online Self-Enhancement Method

---

**Input:** A specific scene set $S$ with a list of objects of interest to users: $List_{in} = [O_1, O_2, ..., O_n, |object, background]$. And the proposal bounding box of Uni-YOLO is denoted as $\Theta(image)$.

1: **for** $i$ in $S$ **do**
2: $\quad [image(i), label(i)] = S(i)$.
3: $\quad dense\_label(i) = \Theta(image(i))$.
4: $\quad p_k = SoftMax(Sim(L(List[k]), V(dense\_label(i))))$.
5: $\quad category(i) = List[Max(p_k)]$.
6: $\quad$ **if** $category(i)$ in $[O_1, O_2, ..., O_n]$ **then**
7: $\quad\quad label(i) = dense\_label(i)$.
8: $\quad$ **end if**
9: $\quad S(i) = [image(i), label(i)]$.
10: $\quad$ Update $\Theta(image)$ with $Loss_{goc}$, $Loss_r$ and $Loss_{con}$ on $S(i)$.
11: **end for**

**Output:** The self-enhanced $\Theta(image)$. It will focus on specific objects of interest even more, in specific scenes.

---

## 3.4 Online Self-Enhancement Method

The trained Uni-YOLO can detect a vast array of categories, but when applied to a given scene, it is not necessary to focus on all categories. Our goal is to enhance the detection of given objects while minimizing the detection of irrelevant ones, resulting in a more adaptable and universal detection system for specific scenes. We propose an online self-enhancement method to improve detection performance for given objects in any given scene. As shown in the right part of Figure 4, the process begins with performing inference in a specific scene, generating dense detection results for various objects. Concurrently, a list of candidate objects of interest, denoted as $[O_1, O_2, ..., O_n, |object, background]$, is constructed. Then, guided by the vision-language model CLIP, the dense results are filtered to select only the objects of interest contained in $[O_1, O_2, ..., O_n]$. The selected objects serve as pseudo-labels for further online fine-tuning of our Uni-YOLO to achieve better detection. The specific steps of this proposed method are summarized in Algorithm 2.

## 3.5 Multimedia UAV Detection Platform

We develop a multimedia interaction UAV platform for object detection as shown in Figure 5. The system's inputs include the user's voice and real-time images captured by the visual sensors. Users can specify objects of interest to the system via voice commands. The voice is then converted into a list of candidate objects using Whisper JAX, a real-time Acoustic-to-Text model [32]. Subsequently, the text embeddings of the candidate objects are obtained from the text encoder of CLIP, and the input images are processed by our Uni-YOLO to obtain detection results in the open world. The detection platform includes the DJI MATRICE M300 RTK as the base UAV, complemented by the MATRICE 350 RTK serving as the human-machine interactive remote controller. The primary imaging sensor used in our UAV system is the ZENMUSE H20 T camera. To process the information collected by the UAV and perform real-time detection, we use the DJI MANIFOLD 2 as the main processor. This computation platform is equipped with both a CPU and GPU, providing the necessary computing power for our object detector Uni-YOLO. The specified computation platform has a theoretical

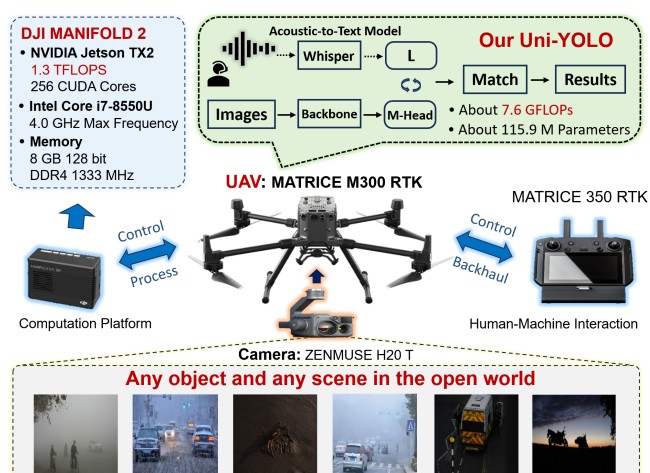

**Figure 5: Illustration for the developed multimedia interaction UAV platform for open-world object detection.**

**Table 1: Large-scale detection datasets used for training.**

| Datasets | Images | Boxes | Categories | Annotation |
|---|---|---|---|---|
| Object365 [36] | 638k | 10,101k | 365 | Dense |
| Pascal VOC [5] | 11.5k | 27k | 20 | Usual |
| COCO [21] | 123k | 896k | 80 | Usual |
| OpenImages [13] | 1,515k | 14,815k | 600 | Sparse |

capacity of 1.3 TFLOPS, which fulfills the 7.6 GFLOPs required by Uni-YOLO. More details about the computation platform and the UAV system can be found on the DJI website[2].

## 4 EXPERIMENTS

### 4.1 Implementation Details

Uni-YOLO is trained on various large-scale object detection datasets, as summarized in Table 1. We conduct training and inference using PyTorch 1.8.1 on an i9-13900K CPU and an NVIDIA 4090 GPU. Details of the UAV system configuration are provided in subsection 3.5. We employ an SGD optimizer with a learning rate of 0.01. The Enhancement Module (EHM) is pre-trained on the scattering dataset RESIDE [15] and the low illuminance dataset LOL [45].

We first evaluate Uni-YOLO in the open world using the low illuminance dataset ExDark [25] and the scattering dataset RTTS [15] to test its generalization and robustness under harsh weather conditions. Additionally, we use the 13 ODinW datasets [17] to further evaluate generalization performance across various complex scenes. Comparisons with existing ov-detectors are also conducted on the OV-COCO dataset [50] in normal environments. Detection performance is measured using mean Average Precision (mAP).

### 4.2 Detection Performance in the Open World

#### 4.2.1 *Based on Scattering and Low Illuminance Dateset.* This experiment demonstrates the zero-shot generalization and robustness of Uni-YOLO in harsh conditions. We compare Uni-YOLO with other ov-detectors, all methods utilizing the CLIP model with an

---

[2]The DJI MANIFOLD 2 website: https://www.dji.com/cn/manifold-2.

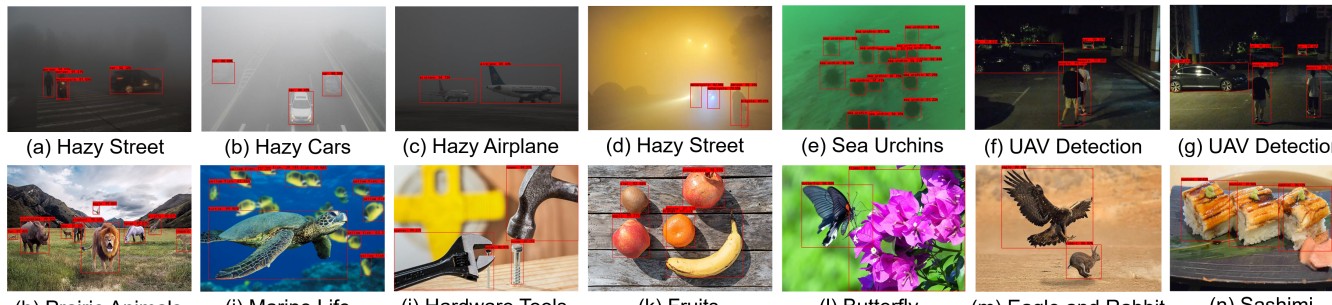

(a) Hazy Street  (b) Hazy Cars  (c) Hazy Airplane  (d) Hazy Street  (e) Sea Urchins  (f) UAV Detection  (g) UAV Detection

(h) Prairie Animals  (i) Marine Life  (j) Hardware Tools  (k) Fruits  (l) Butterfly  (m) Eagle and Rabbit  (n) Sashimi

**Figure 6: Illustration for the zero-shot object detection performance of Uni-YOLO in various scenes. Figure (a)~Figure (e) are various low illuminance or scattering scenes. Figure (f) and Figure (g) are the detection performance of our UAV detection system in a real-world low illuminance scene. Figure (h)~Figure (n) are various scenes with normal illuminance.**

**Table 2: The evaluation results (mAP50 %) for Uni-YOLO compared to other SOTA open-vocabulary detectors, on the low illuminance ExDark [25] and scattering RTTS [15] datasets.**

| Methods | ExDark [25] ↑ | RTTS [15] ↑ |
|---|---|---|
| RegionCLIP (2022 CVPR) [57] | 43.3 | 41.2 |
| OV-DETR (2022 ECCV) [50] | 41.5 | 43.5 |
| PromptDet (2022 ECCV) [7] | 42.4 | 41.9 |
| Detic (2022 ECCV) [58] | 43.5 | 42.4 |
| BARON (2023 CVPR) [46] | 44.1 | **45.4** |
| CORA (2023 CVPR) [47] | 43.2 | 40.5 |
| UniDetector (2023 CVPR) [44] | **44.4** | 44.1 |
| Uni-YOLO (w/o EHM) | 45.1 | 46.4 |
| Uni-YOLO (with EHM) | **53.4** | **52.5** |

**Table 3: The evaluation results (mAP50 %) for the proposed Uni-YOLO (with EHM) compared to other universal object detectors, on the open world 13 ODinW datasets [17].**

| Methods | Training Datasets | mAP ↑ |
|---|---|---|
| GCLIP-T (A) [18] | Object365 | 28.8 |
| GCLIP-T (B) [18] | Object365 | **33.2** |
| UniDetector [44] | Object365 | 30.1 |
| Uni-YOLO | Object365 | 30.6 |
| Uni-YOLO (with S.E.) | Object365 | **37.6** |
| UniDetector [44] | Object 365, COCO, Open Image | **47.3** |
| Uni-YOLO | Object 365, COCO, Open Image | 39.4 |
| Uni-YOLO (with S.E.) | Object 365, COCO, Open Image | **48.2** |

1. The detection results of comparisons are reported by Uni-Detector [44].

**Table 4: The evaluation results (mAP50 %) and Real-Time performance for Uni-YOLO (with EHM) compared to other open-vocabulary detectors, on the OV-COCO dataset [50].**

| Methods | Novel ↑ | Base ↑ | All ↑ | Time(ms) ↓ |
|---|---|---|---|---|
| RegionCLIP (2022 CVPR) [57] | 31.4 | 57.1 | 50.4 | 218 |
| OV-DETR (2022 ECCV) [50] | 29.4 | 61.0 | 52.7 | **148** |
| PromptDet (2022 ECCV) [7] | 26.6 | 59.1 | 50.6 | 256 |
| Detic (2022 ECCV) [58] | 27.8 | 47.1 | 42.1 | 217 |
| BARON (2023 CVPR) [46] | 34.0 | 60.4 | 53.5 | 212 |
| CORA (2023 CVPR) [47] | 35.1 | 35.5 | 35.4 | 156 |
| UniDetector (2023 CVPR) [44] | **35.2** | 56.8 | 51.2 | 202 |
| Uni-YOLO | **36.6** | 54.8 | 50.1 | **33** |

1. To ensure fairness, all comparison methods are based on CLIP with the RN50 backbone only, without additional caption supervision.

best by 7.1%. Some examples of zero-shot detection performance in harsh conditions are shown in Figure 6 (a)~(d), with detection performance in scattering underwater shown in Figure 6 (e).

*4.2.2* ***Based on the 13 ODinW Datasets.*** This experiment demonstrates the generalization of Uni-YOLO in the complex open world. We perform comparison experiments with other universal detectors on the 13 ODinW datasets [17]. The datasets contain thirteen subsets and have various scenes to simulate the complex open world. The Table 3 provides a summary of the test results. The results show that Uni-YOLO has superior universal detection performance in the open world, achieving 37.6% mAP based on training with only Object365 dataset, which beats the best previous method (33.2%) by 4.4%. When we use more datasets to perform training, Uni-YOLO achieves a further improvement, achieving 48.2% mAP, which beats the best previous method (47.3%) by 0.9%. Some examples of detection performance in various scenes are shown in Figure 6 (h)~(n).

*4.2.3* ***Based on the UAV Detection System.*** This experiment demonstrates the practical value of Uni-YOLO in the real world. We employ the developed multimedia interaction UAV platform for real-world testing in a nighttime street setting. The candidate objects of interest to the user are "people" and "cars". Detection is performed at up to 20 FPS on the Jetson TX2 GPU. More real-world

RN50 backbone and without additional caption supervision. We perform comparison experiments with other ov-detectors on the low illuminance dataset ExDark [25] and the scattering dataset RTTS [15]. A summary of the test results can be found in Table 2. Uni-YOLO achieves 53.4% mAP on ExDark, surpassing the previous best by 9.0%, and 52.5% mAP on RTTS, outperforming the previous

**Table 5: The test results for the ablation studies of the proposed object augmentation method (Algorithm 1) based on the 13 ODinW datasets [17]. We use two methods to train our Uni-YOLO based on the multiple source datasets. The *Compound* method simply mixes these datasets sequentially. The *Augment* denotes the use of the object augmentation.**

| Multiple Source Datasets | | All mAP | S.fish | VOC | Drone | Aq.ium | Rabbit | EGO | M.room | Package | Raccoon | Vehicle | Pistol | Therm. | Poth. |
|---|---|---|---|---|---|---|---|---|---|---|---|---|---|---|---|
| A: [$Obj.365$] | – | **30.6** - | 15.6 | 36.9 | 18.7 | 27.1 | 86.0 | 3.7 | 21.0 | 54.6 | 79.0 | 43.3 | 10.3 | 1.9 | 0.3 |
| B: [$A, VOC$] | Compound | **33.1** - | 16.7 | 44.5 | 21.6 | 26.3 | 85.0 | 3.6 | 19.4 | 62.9 | 84.4 | 51.0 | 12.8 | 2.0 | 0.7 |
| | Augment | **32.4** ↓ | 16.1 | 43.6 | 21.6 | 26.7 | 87.0 | 1.9 | 21.0 | 62.9 | 85.4 | 37.2 | 14.2 | 2.9 | 0.9 |
| C: [$B, CO$] | Compound | **32.4** - | 17.9 | 40.5 | 20.0 | 28.6 | 88.5 | 2.4 | 14.6 | 65.3 | 80.0 | 47.4 | 12.2 | 2.4 | 0.8 |
| | Augment | **34.9** ↑ | 16.5 | 45.6 | 22.6 | 29.7 | 88.9 | 4.4 | 22.2 | 75.2 | 88.7 | 39.7 | 16.2 | 3.5 | 1.4 |
| [$C, O.Image$] | Compound | **33.7** - | 14.6 | 39.1 | 18.9 | 30.9 | 87.3 | 2.4 | 18.9 | 69.1 | 81.4 | 53.6 | 18.9 | 2.4 | 1.1 |
| | Augment | **39.2** ↑ | 20.0 | 47.3 | 22.6 | 36.9 | 91.0 | 13.3 | 27.3 | 79.2 | 86.6 | 51.8 | 24.1 | 6.7 | 2.6 |

**Table 6: The test results for the ablation study of the self-enhancement in scattering scenes. We perform self-enhancement for five objects based on the RTTS dataset [15].**

| Methods | Person | Bicycle | Car | Bus | Motorbike |
|---|---|---|---|---|---|
| *Initial Uni-YOLO* | **68.9** - | **45.2** - | **66.0** - | **43.9** - | **38.5** - |
| S.E. Person | **73.2** ↑ | 1.8 | 0.1 | 0.1 | 6.1 |
| S.E. Bicycle | 9.7 | **49.6** ↑ | 0.5 | 0.9 | 10.1 |
| S.E. Car | 0.3 | 0.5 | **69.7** ↑ | 6.1 | 1.6 |
| S.E. Bus | 2.7 | 4.3 | 15.3 | **52.1** ↑ | 5.0 |
| S.E. Motorbike | 5.7 | 11.5 | 1.3 | 0.3 | **43.4** ↑ |

detection results and visual representations of the developed UAV detection system are provided in the supplementary material.

## 4.3 Comparison with Open-Vocabulary Methods

This experiment demonstrates the zero-shot performance of Uni-YOLO in normal environments, and its real-time performance. The common public benchmark OV-COCO [50] is used for evaluation, containing 17 novel and 48 base categories. Consistent with the previous methods, the base categories are used for training and the novel categories are used for zero-shot testing. A summary of the test results can be found in Table 4. The results demonstrate that Uni-YOLO also has superior zero-shot performance in normal environments, achieving 36.6% mAP for novel categories, which exceeds the best previous method (35.2%) by 1.4%. The results also show that Uni-YOLO provides superior real-time performance (33 ms per image, about 30 FPS). Since it is designed as a single-stage, it has an obvious real-time superiority to other two-stage methods.

## 4.4 Ablation Studies

*4.4.1 **Object Augmentation Training Method**.* This experiment evaluates the proposed object augmentation training method, based on 13 ODinW datasets [17]. First, Uni-YOLO is trained on the relatively densely annotated dataset, Object365 [36]. Two methods are used to introduce additional datasets for training. The *Compound* method involves the simple sequential blending of these datasets, the *Augment* method uses the proposed object augmentation (Algorithm 1). We gradually include the Pascal VOC [5], COCO [21], and OpenImages [13] datasets for further training. The results of the experiments are summarized in Table 5. In most cases, *Augment*-based training produces better results than the simple *Compound* method. On average, the *Augment*-based multi-sources data

training achieves a 39.2% mAP, representing an 8.6% increase over using the single Object365 dataset, while the Compound method only achieves a 33.7% mAP. The results demonstrate the importance of using multi-source data and the effectiveness of using the consistent annotations provided by the proposed augmentation.

*4.4.2 **Online Self-Enhancement Method**.* The above test results in Table 3 demonstrate the effectiveness of self-enhancement (***S.E.***), achieving 8.8% (from 39.4% to 48.2%) and 7.0% (from 30.6% to 37.6%) improvement. Additionally, we sequentially perform five categories self-enhanced (***S.E.***) based on the RTTS dataset [15] to evaluate the effectiveness of the method in scattering scenes. The results are summarized in Table 6. The results show that self-enhancement method enables Uni-YOLO to improve the detection performance of given objects. We observe a sequential improvement in the detection accuracy for the five categories. Thus, based on this method, Uni-YOLO can improve the detection of specific objects in specific scenes by itself, without human annotation.

*4.4.3 **Enhancement Module**.* This experiment evaluates the proposed enhancement module (EHM). Table 2 provides the experimental results of the two types of Uni-YOLO, with EHM and without EHM. The detection performance is improved with the EHM under harsh conditions, achieving 53.4% mAP on the low illuminance dataset, which outperforms the without EHM method (45.1%) by 8.3%, and achieving 52.5% mAP on the scattering dataset, which outperforms the without EHM method (46.4%) by 6.1%. Based on the proposed EHM, the robustness of Uni-YOLO's detection performance under different weather conditions is effectively improved.

## 5 CONCLUSIONS

In this paper, we propose Uni-YOLO, a robust and fast universal object detector. It is a new one-stage detector for object detection in the open world. Uni-YOLO utilizes general object confidence to effectively distinguish between objects and backgrounds, incorporating a grid cell method for precise bounding box regression. We design a physical model-based EHM to provide adaptive enhancement for Uni-YOLO in harsh weather conditions. We also propose the object augmentation method to train Uni-YOLO and design the self-enhancement method to online fine-tune Uni-YOLO. Comprehensive experiments on public benchmarks and the deployment of a UAV demonstrate its real-time robust detection performance.

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
