# OpenReview forum: "Uni-YOLO: Vision-Language Model-Guided YOLO for Robust and Fast Universal Detection in the Open World"
_acmmm.org/ACMMM/2024/Conference — MM2024 Poster_

### Official Review · Reviewer_mXMh · 2024-05-05

**Rating:** 3
**Confidence:** 4

**Summary:**

This paper presents a new single-stage generalized object detector. it employs a generalized target confidence level to discriminate between target and background, and uses a grid cell regression approach to achieve real-time detection. In the article, EHM is designed to enhance the robustness of Uni-YOLO under adverse weather conditions. During training and inference, Uni-YOLO is guided by a large-scale visual language model, CLIP.. In addition, the article proposes an online self-enhancement method that allows Uni-YOLO to further focus on specific targets in a given scene through self-supervised fine-tuning. The superiority and practical value of the method are verified in public benchmarking and UAV deployment experiments.

**Strengths:**

1. The paper is written in a logical and clear manner, with a clear hierarchy and visual presentation of graphs and charts. It stimulates the reader's interest in the study.
2. The proposed one-stage Uni-YOLO architecture with parallel detection heads and EHM provides a novel and efficient approach for open-world object detection under various conditions.
3. The object augmentation method addresses the important problem of inconsistent annotations across source datasets, enabling effective large-scale training.
4. The paper includes thorough experiments on a variety of datasets (ExDark, RTTS, ODinW, OV-COCO) and real-world UAV testing, providing strong empirical support for Uni-YOLO's effectiveness and practical value.

**Limitations:**

Weaknesses:
1. There are concerns regarding the fairness of the comparative experiments. The proposed method utilizes EfficientNet for feature extraction, yet it does not specify the backbone used by the comparative methods. Additionally, the comparison does not appear to be conducted using the same benchmark detectors, such as the MaskRCNN utilized by GLIP.
2. A significant enhancement in the performance of the proposed method is contingent upon the EHM and the S.E. strategy. These components seem applicable to other open-vocabulary detectors as well. However, the paper does not validate their potential as a general contribution to the field.
3. The paper does not mention or compare with other ov and single-stage object detection works, which is a notable omission considering the relevance to the study's focus.

[1] Open-Vocabulary One-Stage Detection with Hierarchical Visual-Language Knowledge Distillation. CVPR2022.

[2] EdaDet: Open-Vocabulary Object Detection Using Early Dense Alignment. ICCV 2023.

[3] Zero-shot object detection through vision-language embedding alignment. ICDMW 2022.

4. In comparison experiments with other state-of-the-art open vocabulary detectors (Table 4), the real-time performance of Uni-YOLO (33 ms) is much better than previous methods (e.g., OV-DETR 148 ms, CORA 156 ms, etc.). I encourage the authors to explain the reasons.

Questions:

1. How was the vocabulary list for zero-shot classification (Section 3.1) constructed in the experiments - manually or automatically? Are there any limitations or sensitivities around this?
2. What criteria or thresholds are used to select the pseudo-labels from CLIP during object augmentation and self-enhancement? How do these impact precision/recall?
3. Can the EHM architecture and training process be elaborated? E.g. what datasets were used, how were the branches designed?

I will revise my rating based on other comments and the author's rebuttals.

**Suitability:**

2

---

### Official Review · Reviewer_wuAn · 2024-05-23

**Rating:** 4
**Confidence:** 3

**Summary:**

This paper introduces a one-stage detector for object detection in the open world. Uni-YOLO utilizes general object confidence to effectively distinguish between objects and backgrounds, incorporating a grid cell method for precise bounding box regression. The authors also design a physical model-based EHM to provide adaptive enhancement for Uni-YOLO in harsh weather conditions.

**Strengths:**

The concept of Universal Detection in the Open World is practical and reasonable. The authors have considered both Harsh Weather and Open Vocabulary settings, and the experimental results are impressive.

**Limitations:**

1-Although the paper presents excellent work, the impact is somewhat limited by the fact that the code and models for Uni-YOLO are not open-sourced, especially considering that most of the compared methods (e.g., UniDetector, CORA, BARON, Detic) have released their code.

2-To demonstrate the adaptability of the proposed method to harsh weather, it would be better to include comparisons with specific harsh weather detectors (Table 2 seems to include only general detectors).

3-In this work, harsh weather conditions appear to be limited to poor lighting scenarios (using EHM to enhance images). How does the method handle other conditions such as rain or snow?

4-In Table 4, Uni-YOLO does not show an advantage over previous works in the "Base" and "All" settings. What could be the possible reasons for this?

**Suitability:**

3

---

### Official Review · Reviewer_ek2D · 2024-05-29

**Rating:** 4
**Confidence:** 4

**Summary:**

This paper develops Uni-YOLO, a robust and fast detector that would deal with open-vocabulary objects in challenging weather. It develops several techniques to improve the robustness of the model, e.g., input enhancement, object augmentation, etc. Uni-YOLO achieves good performance on several benchmarks with fast inference speed.

**Strengths:**

1. The proposed method reaches a decent balance between speed and performance across different benchmarks.
2. This paper is clearly written, and the figures and supplementary videos are well demonstrated, making this paper easy to understand.
3. Some proposed novel modules are effective like object augmentation. It makes sense to deal with the annotation conflicts among different datasets when conducting joint training.

**Limitations:**

1. The novelty is limited.
The proposed modules are straightforward combinations of existing techniques.  Like the open-vocabulary pipeline design and the object augmentation methods.

2. Lacks ablation studies.
Could the input enhancement module be replaced by simple data augmentations?
The effectiveness of the object score is not verified.
A step-by-step ablation from the naive baseline to the full-version model is expected. We could not figure out where does the improvement come from.

3. Lacks discussions for similar works.
The author should add [A1]  for discussion and comparisons.

[A1]YOLO-World: Real-Time Open-Vocabulary Object Detection

**Suitability:**

2

---

### Official Review · Reviewer_5MFw · 2024-06-02

**Rating:** 5
**Confidence:** 4

**Summary:**

This paper proposes Uni-YOLO, a universal detector designed for complex scenes with real-time performance.

**Strengths:**

1. This paper studies universal object detection in harsh weather, which has not been studied in existing works.
2. This paper realizes real-time object detection.
3. The experimental results are good.

**Limitations:**

1. The article proposes EHM to overcome harsh weather conditions, mainly on two aspects: scattering and illuminance. However, harsh weather contains a wide variety of conditions. Restricting the discussion of harsh weather to these two aspects may limit the method's universal ability in harsh weather scenarios. The authors should conduct experiments under more diverse harsh weather conditions and compare with previous approaches to demonstrate its universality in handling harsh weather.
2. Table 2 should specify the training data used for each method. It seems that the training data for these methods is not consistent.
3. Since it focuses on real-time detection, I recommend that the authors include a detailed table about model efficiency. This table should compare metrics such as GFLOPs, parameters, and FPS with existing methods.
4. The ablation study should be more detailed. About EHM, the authors should separately discuss the impact of scattering degradation and illuminance degradation on the results.

**Suitability:**

3

---

### Meta-Review · Area_Chair_aqFq · 2024-07-03

**Recommendation:** Accept (Poster)
**Confidence:** 5

**Metareview:**

This paper was reviewed by four experts in the field. The recommendations are (Weak Accept, Borderline Accept x 3). Based on the reviewers' feedback, the decision is to recommend the acceptance of the paper. The reviewers did raise some valuable concerns (especially more detailed ablations experiments, numerical results for model efficiency, more comparisons with related works, etc.) that should be addressed in the final camera-ready version of the paper. The authors are encouraged to make the necessary changes to the best of their ability. We congratulate the authors on the acceptance of their paper.